# Photosynthesis Monitoring in Microalgae Cultures Grown on Municipal Wastewater as a Nutrient Source in Large-Scale Outdoor Bioreactors

**DOI:** 10.3390/biology11101380

**Published:** 2022-09-22

**Authors:** Jiří Masojídek, Cintia Gómez-Serrano, Karolína Ranglová, Bernardo Cicchi, Ángel Encinas Bogeat, João A. Câmara Manoel, Ana Sanches Zurano, Ana Margarita Silva Benavides, Marta Barceló-Villalobos, Victor A. Robles Carnero, Vince Ördög, Juan Luis Gómez Pinchetti, Lajos Vörös, Zouhayr Arbib, Frank Rogalla, Giuseppe Torzillo, Félix Lopez Figueroa, Francisco Gabriel Acién-Fernándéz

**Affiliations:** 1Laboratory of Algal Biotechnology, Centre ALGATECH, Institute of Microbiology, Czech Academy of Sciences, 37901 Třeboň, Czech Republic; 2Faculty of Science, University of South Bohemia, 37005 České Budějovice, Czech Republic; 3Department of Chemical Engineering, University of Almería, 04120 Almería, Spain; 4CIESOL and Solar Energy Research Centre, Joint Centre University of Almería-CIEMAT, 04120 Almería, Spain; 5CNR—Institute of Bioeconomy, 50019 Sesto Fiorentino, Italy; 6Departamento de Innovación y Tecnología, Aqualia, 28231 Mérida, Spain; 7CIMAR—Centro de Investigación en Ciencias del Mar y Limnología, Universidad de Costa Rica, San Pedro 2060, Costa Rica; 8Department of Ecology, Institute of Blue Biotechnology and Development (IBYDA), University of Malaga, 29071 Málaga, Spain; 9Department of Plant Science, Faculty of Agricultural and Food Sciences, Széchenyi István University, 9200 Mosonmagyaróvár, Hungary; 10Spanish Bank of Algae (BEA), Instituto de Oceanografía y Cambio Global (IOCAG), Universidad de Las Palmas de Gran Canaria, 35214 Telde, Spain; 11Balaton Limnological Research Institute, 8237 Tihany, Hungary

**Keywords:** biomass, biostimulanting activity, chlorophyll fluorescence, microalga, *Micractinium*, oxygen production, photosynthesis, raceway pond, large-scale bioreactor, wastewater

## Abstract

**Simple Summary:**

The remediation of waste nutrients, mainly nitrogen, phosphorous, and carbon, results in low-cost biomass production. In this work, raceway ponds with an area of 1 and 0.5 ha were built in Mérida, Spain next to a municipal wastewater (WW) treatment plant. These DEMO bioreactors are operated continuously all year round. At the start of the trials, the bioreactors were filled with urban WW as a sole source of nutrients and left to be populated by the sewage-born phytoplankton. The fitness and growth of the phytoplankton population (dominated by the green microalga *Micractinium*) were monitored by photosynthesis measuring techniques. The cultures showed suitable photosynthetic activity. Productivity values ranged between 15 and 25 g DW m^−2^ day^−1^ and processed biomass revealed biostimulating activity. In these trials, photosynthesis measuring techniques, i.e., oxygen evolution and chlorophyll (Chl) fluorescence techniques, were validated to monitor large-scale bioreactors using municipal WW remediation for biomass production, which can be used for agricultural purposes as biofertilizer/biostimulant.

**Abstract:**

Microalgae cultures were used for a WW treatment to remediate nutrients while producing biomass and recycling water. In these trials, raceway ponds (RWPs; 1 and 0.5 ha) were located next to a municipal (WW) treatment plant in Mérida, Spain. The ponds were used for continuous, all-year-round microalgae production using WW as a source of nutrients. Neither CO_2_ nor air was supplied to cultures. The objective was to validate photosynthesis monitoring techniques in large-scale bioreactors. Various in-situ/ex-situ methods based on chlorophyll fluorescence and oxygen evolution measurements were used to follow culture performance. Photosynthesis variables gathered with these techniques were compared to the physiological behavior and growth of cultures. Good photosynthetic activity was indicated by the build-up of dissolved oxygen concentration up to 380% saturation, high photochemical yield (Fv/Fm = 0.62–0.71), and relative electron transport rate rETR between 200 and 450 μmol e^−^ m^−2^ s^−1^ at midday, which resulted in biomass productivity of about 15–25 g DW m^−2^ day^−1^. The variables represent reliable markers reflecting the physiological status of microalgae cultures. Using waste nutrients, the biomass production cost can be significantly decreased for abundant biomass production in large-scale bioreactors, which can be exploited for agricultural purposes.

## 1. Introduction

The high adaptability of microalgae (prokaryotic cyanobacteria and eukaryotic algae) allows them to grow in a wide variety of environments, including municipal, industrial, and agricultural wastewater (WW). Oswald et al. [1] were the first to describe this technology and microalgae-bacteria interactions in WW treatment. Since the 1960s microalgae have been tested and applied to WWT due to their ability to recuperate inorganic and organic nutrients, e.g., carbon, nitrogen, phosphorous, as well as some other compounds, such as pharmaceuticals or heavy metals, while reducing chemical oxygen demand (COD) and producing biomass [2,3,4,5,6,7]. Municipal WW is an abundant source of nutrients (nitrogen, phosphorus, and organic carbon) as their total production increases proportionally to the rise of the human population. At present, microalgae cultivation on WW has been one of the most promising and sustainable approaches in the perspective of circular economy. 

Applying microalgae to WWT associate them with other aerobic and anaerobic microorganisms. Thus, the microalgae–bacteria consortia in engineered bioreactors represent a platform for WWT based on the combination of phototrophic and heterotrophic cultivation regimes [8,9]. In the WWT process, bacteria are responsible for chemical oxygen demand (COD) used for the degradation of organic material into mineral components, while using photosynthetic oxygen and releasing carbon dioxide by respiration. On the contrary, microalgae consume CO_2_ and mineral nutrients to produce biomass and oxygen required by bacteria [10]. These procedures represent a cost-effective and sustainable process based on solar-driven oxygenation (mediated by photosynthesis), enhanced nutrient assimilation (as a result of the combined heterotrophic and phototrophic growth), and efficient pathogen removal (due to the high pH and oxygen concentrations mediated by photosynthesis) [8,10]. 

Up to now, the commercial production of microalgae biomass has been realized on a scale of hectares producing thousands of metric tons. Recently, numerous studies have been published that focus on the use of microalgae for biofuel production [11,12,13], or aquaculture and agricultural applications which may demand ample production [14,15,16,17]. However, to satisfy these markets, the cost of biomass production must be significantly reduced while the production capacity has to be substantially increased to hundreds of thousands of tons [4,18]. The production of microalgae biomass in such amounts has to be managed in large-scale units, bioreactors. The so-called raceway ponds (RWPs) have been commonly used for microalgae mass production [8,18,19,20,21]. These systems are constructed as a shallow loop channel in which the culture is continuously mixed by paddle wheels, thus improving the growth of algaculture compared to natural reservoirs [18]. 

Currently, raceway ponds have also been the most widespread cultivation system used for WWT and they are considered one of the most cost-efficient to produce microalgae biomass [18,22]. This production in combination with WWT has been applied only recently in Europe on a larger commercial scale of hectares intended for biofuel production, for example, a 10-ha plant operated by Aqualia in Chiclana de la Frontera, Spain [23]. Recently, two DEMO plants, each of about 2 and 3 ha, were built in Mérida and Hellín (Spain) to produce biomass for aquaculture and agriculture use (project H2020 Sabana; https://www.eu-sabana.eu (accessed on 18 August 2022)). 

Usually, there are no specific strains uniquely suited for WWT. Thus, often robust species locally isolated, having high biomass productivity and tolerance to varying outdoor conditions are being tested. The strains most frequently employed in WWT processes are Chlorophyceae (*Chlorella*, *Scenedesmus*, *Micractinium*, *Muriellopsis*, etc.) as well as some cyanobacteria (*Nostocales*) [24,25]. Some of these strains synthesize an array of secondary high-value metabolites with biological activities, e.g., phytohormones that can be used as plant biostimulants in agriculture [17]. 

In this work, photosynthesis measuring techniques were validated for monitoring microalgae cultures grown in outdoor bioreactors which were used for municipal WW remediation. Photosynthetic activity measured by oxygen production and Chl fluorescence, in-situ/ex-situ, was found as a suitable tool to follow the physiological status and can be used to estimate the growth regime of microalgae cultures in large-scale bioreactors.

## 2. Materials and Methods 

### 2.1. Cultivation Plant

Large-scale raceway bioreactors were built in the DEMO plant in Mérida, Spain (GPS coordinates 38°55′52.1″ N, 6°23′35.7″ W) by the company Aqualia: RWP1 with an area of 10,000 m^2^ (volume of about 3000 m^3^), and RWP2 with an area of 5000 m^2^ (volume of about 1250 m^3^). In the bioreactors, average depths were about 27 cm with a culture flow rate of about 0.3 m s^−1^ (Figure 1A). The bioreactors were installed next to a WW treatment plant (Estación de Depuración de Aguas Residuales, Mérida). The microalgae cultures in bioreactors were circulated by rotating paddle wheels; flow deflectors were placed at both ends (Figure 1). The trials were carried out in mid-August 2021. Except for ambient, neither additional CO_2_ nor air was available in the bioreactors. 

Microalgae cultures were grown in municipal WW from local WWTP as it was the only source of nutrients. The flow rate of incoming WW was about 25 m^3^ h^−1^ in RWP1 and about 10 m^3^ h^−1^ in RWP2. The WW samples were collected directly from the inlet tube without filtration. No nutrients were added to bioreactors. The parameters of the incoming WW are shown in Table 1. The analyzed value of chemical oxygen demand (COD) was about 550 mg DW L^−1^ and the determined amount of total suspended solids (TSS) was about 100–150 mg DW L^−1^. The content of the macronutrients, nitrogen and phosphorus was 58 and 10.7 mg DW L^−1^, respectively, which is about 10-times lower compared to commonly used laboratory cultivation media such as BG−11 [26]. The output culture samples were collected from the bioreactor overflow and filtered to remove cells.

### 2.2. Photosynthesis Monitoring 

Photosynthetic activity of microalgae populations was monitored by Chl fluorescence and oxygen production measurements in the bioreactors in-situ as well as ex-situ in culture samples taken from outdoor cultures. The daytime in measurements corresponds to CEST (GMT + 1).

#### 2.2.1. In-Situ Measurements

Chl fluorescence data was recorded in-situ using a portable fluorimeter (Junior-PAM, H. Walz GmbH, Effeltrich, Germany) controlled by the WinControl−3.2 software via a USB interface which was also used for data acquisition [27,28]. The fluorimeter was fitted with blue light-emitting diodes (LED, 460 nm) to apply the measuring as well as saturating pulses. In outdoor measurements, ambient irradiance was used as actinic light. The measuring light guide (optical plastic fiber; 1.5 mm in diameter and 100 cm long) and a spherical PAR mini-sensor (US-SQS, H. Walz GmbH, Effeltrich, Germany) were submerged next to each other in the middle of the culture photic zone (at a depth of about 15 mm) to measure photosynthesis variables. The incident photosynthetically active radiation E_PAR_ (400–700 nm) and the actual quantum yield of PSII [Y(II)~ΔF′/Fm′ = (Fm′ − F′)/Fm′] were measured each 10 min in the outdoor culture where the variable F′ is the steady-state fluorescence level and Fm′ and Fm are the maximum fluorescence yields induced by a saturating light pulse which was measured in the light- and dark-adapted cultures, respectively [29]. The relative electron transport rate rETR [= Y(II) × E_PAR_] through PSII (µmol electrons m^−2^ s^−1^) was used to estimate photosynthetic activity, where E_PAR_ is the incident photosynthetically active irradiance (µmol photons m^−2^ s^−1^). This variable is easy to record and adequate to follow the diurnal changes in physiological conditions in outdoor microalgae cultures. It is important to point out that rETR represents a relative value, and therefore it should be used as a variable to compare the performance of cultures within the same trial or unit.

Temperature, pH values, and dissolved oxygen (DO) concentrations were measured by a hand-held pH-meter (pH3310, WTW, Weilheim, Germany) and an oximeter with temperature compensation (model Oxi330, WTW, Weilheim, Germany).

#### 2.2.2. Ex-Situ Measurements

For ex-situ measurements of Chl fluorescence and oxygen production, microalgae samples were taken from outdoor cultures at specified time intervals (9:00, 13:00, and 17:00 h) as described previously [30].

Rapid light response curves (RLC) were recorded using a portable fluorimeter (Junior-PAM, H. Walz, Effeltrich, Germany) using a series of 12 light intensities between 0 and 1500 µmol photons m^−2^ s^−1^ with a 10-s exposure at each intensity. The curves were evaluated and the maximum photochemical efficiency of PSII, Fv/Fm, and the relative electron transport rate rETR were calculated using the WinControl−3 software. The minimum and maximum fluorescence levels F_0_, Fm′, and Fm (F_0_, basal fluorescence from fully oxidized reaction centers of PSII; Fm and Fm′—maximum fluorescence from partially or fully reduced PSII reaction center) were determined in the dark-adapted samples. The maximum PSII quantum yield was calculated as the ratio of variable and maximum fluorescence, Fv/Fm = (Fm − F_0_)/Fm, which indicates the capacity of the system to absorb light through the reaction centers and the light-harvesting complex and expresses the maximum quantum efficiency of primary photochemistry [31,32]. The variable called the relative electron transport rate through PSII (rETR) was calculated as the product of the actual photochemical efficiency Y(II) multiplied by the photosynthetically active radiation rETR [= Y(II) × E_PAR_; in μmol electrons m^−2^ s^−1^] (e.g., [33,34]). The values of ETRmax and Pmax were calculated at the maxima of ETR vs. irradiance (RLC) curves. Non-photochemical quenching NPQ [= (Fm − Fm′)/Fm′] was used to estimate non-photochemical energy dissipation [35].

‘Slow’ (classical) photosynthesis light-response curves (P vs I curve) of electron transport and oxygen production were measured in parallel using a pulse-amplitude-modulation fluorometer (PAM−2100, H. Walz, Effeltrich, Germany) and an oxygen monitoring system (Oxylab+, Hansatech Instr. Ltd., King’s Lynn, UK) connected to a temperature-controlled chamber DW2/2. The curves were recorded in samples taken from outdoor cultures after 10–15 min dark adaptation at 8 light intensities between 0 and 1800 µmol photons m^−2^ s^−1^) exposing them 2 min under each intensity at the temperature set according to the actual culture value. The curves were evaluated to calculate the maximum photochemical efficiency of PSII, Fv/Fm, and the relative electron transport rate rETR using the WinControl−3 software. The values of ETRmax were calculated at the maxima of ETR vs. irradiance curves by Oxylab software.

Rates of dark respiration and photosynthetic oxygen evolution were measured as a function of irradiance in microalgae samples using a Clark-type oxygen electrode placed in a temperature-controlled chamber (DW2/2, Hansatech Instrument Ltd., Norfolk, UK) and connected to the Oxylab control box (Hansatech Instrument Ltd., Norfolk, UK). The irradiance source was programmed by the O2View software to increase the intensity in 6 two-min steps between 0 and 950 µmol photons m^−2^ s^−1^ (saturating intensity for photosynthesis). The maximum values of respiration and photosynthetic activity Pmax were calculated from photosynthesis light-response curves. The values of respiration and oxygen production are expressed in μmol O_2_ mg^−1^ Chl h^−1^.

#### 2.2.3. Fast Fluorescence Induction Kinetics (Kautsky Curve or OJIP Test)

While the pulse-amplitude-modulation (PAM) technique provides information on the energy distribution between the photochemical and non-photochemical processes in photosynthesis, the fast fluorescence induction kinetics gives information on the redox status of the electron transport chain in the PSII complex. The fluorescence induction curves were measured ex-situ using a portable fluorimeter (AquaPen AP−100, P.S.I. Ltd., Brno, Czech Republic) in samples taken from outdoor cultures, dark-adapted for 5–10 min as described previously [27]. The fast Chl fluorescence induction kinetics (also called the Kautsky curve) of all oxygenic photosynthetic organisms is characterized by a polyphasic rise when the signal rises rapidly from the origin (F_0_ = O) to the highest peak (P = F_M_) via two infections, J and I [36].

The curves of fast fluorescence induction kinetics (OJIP test) were measured in the time range between 50 µs to 1 s when the signal rises rapidly from the origin (O) to the highest peak (P) via two inflections—J and I. The O point (50 µs) of the fluorescence induction curve represents a minimum value (designated as constant fluorescence yield F_0_) when PQ electron acceptors (Q_A_ and Q_B_) of the PSII complex are oxidized. The inflection J occurs after ~2–3 ms of illumination and reflects the dynamic equilibrium (quasi-steady-state) between Q_A_ and Q_A_^−^. The J–I phase (at 30–50 ms) is due to the closure of the remaining centers, and the I–P (ends at about 300–500 ms) corresponds to the full reduction of the plastoquinone pool (equivalent to maximum fluorescence level Fm) [36]. From the fluorescence levels at the J and I points, the variables Vj and Vi were calculated as follows: Vj = (F_2ms_ − F_0_)/(Fm − F_0_) and Vi = (F_30ms_ − F_0_)/(Fm − F_0_), which showed the redox status of quinone electron acceptors.

### 2.3. Biomass Determination and Biochemical Measurements

The measurement of biomass concentration was carried out as dry weight (DW) by filtering culture samples on pre-weighed glass microfiber filters (GC−50) as described previously [37]. The filters with the biomass were washed twice with deionized water and dried in an oven at 105 °C for 3 h. Then, they were weighed (precision of ± 0.01 mg) and the biomass was calculated.

Chlorophyll concentration was determined spectrophotometrically in methanol extracts. The samples were collected by centrifugation and the pellets were suspended in 100% methanol. Sea sand was added, and the tubes were put into the laboratory ultrasound bath for 2 min, then cooled down in an ice bath and centrifuged. The absorbance of the supernatant was measured using a high-resolution spectrophotometer and the concentration of chlorophyll was calculated accordingly [38].

For microscopic observation of culture composition, the samples from each bioreactor were fixed by adding Lugol solution and kept refrigerated (4 ℃) before microscopic analysis.

### 2.4. Determination of Biological Activity

The freeze-dried biomass of microalgae cultures harvested from bioreactors by centrifugation was suspended in distilled water (10 mg DW L^−1^) and sonicated (3 min) for testing biostimulating activities and then diluted to 2 g L^−1^ dry matter concentration. The bioassays were carried out using these extracts. Four different bioassays were used to detect plant biostimulating activities: excised cucumber cotyledon expansion, mung bean rooting, cucumber cotyledon root formation, and leaf chlorophyll retention. All bioassays were performed in triplicate.

#### 2.4.1. Cucumber Cotyledon Expansion Test

The cucumber (*Cucumis sativus* L.) cotyledon expansion bioassay was used to determine the cytokinin-like activity of microalgae strains [39]. For this bioassay, the freeze-dried biomass extracts were diluted to 2 g DW L^−1^ concentration. Cucumber seeds were germinated in the dark in Knopp medium solidified with agar and cotyledons were excised from 5-day-old seedlings. Ten cotyledons were exposed to green light in a Petri dish and treated with the microalgae extracts; the activity was compared with kinetin (6-furfurylaminopurine) solutions as the positive control applied at concentrations of 0.3–1 mg DW L^−1^, and distilled water was used as the negative control. The cotyledons were incubated in the dark for 3 days at 25 °C. The fresh weight of the cotyledons was weighed and compared with the kinetin treatments.

#### 2.4.2. Leaf Chlorophyll Retention Test (Cytokinin-Like Activity)

The cytokinin-like activity was also determined by the leaf chlorophyll retention test [40]. Wheat seeds (*Triticum aestivum* L.) were planted in moistened perlite in plastic trays at 25 °C, about 60 to 65% relative humidity, and 120 µmol photons m^−2^ s^−1^. Leaves from wheat seedlings (about 10 cm in height) were collected and then cut 35 mm below their apical tip into 10 mm segments. The fresh weight of detached leaf segments was weighted and placed in glass vials containing 10 mL of growth medium as the negative control, and microalgae extracts (0.5, 1.0, and 2.0 g DW L^−1^), as treatments. After the 4-day incubation period, the leaf segments were extracted by 80% ethanol at 80–90 °C for 10 min and the chlorophyll extract was then assayed at 645 nm. A standard curve of kinetin (KIN) as the positive control at the concentrations of 0.3–1 mg DW L^−1^ was used for comparison.

#### 2.4.3. Mung Bean Rooting Test

Water extracts of the freeze-dried microalgae biomass (3 mg DW L^−1^) were prepared by 3 min sonication. The plant biostimulating activity of the extracts was measured using the mung bean (*Vigna radiata*) rooting bioassay [41]. The beans were germinated in moist vermiculite at 26 °C in a 16:8 h light:dark photoperiod and 120 μmol photons m^−2^ s^−1^ light intensity. On day 10, uniform mung bean cuttings with two leaves were placed in the growth chamber in the vials with the microalgae extracts (0.5, 1.0, 2.0, and 3 g DW L^−1^) as the treatment for 6 h and transferred to vials containing water. Distilled water was used as the negative control. Then, the plants were put back in the growth chamber for 8 days. After the incubation period, the number of roots (longer than 1 mm) was counted on each hypocotyl. The mean numbers of roots were compared to a standard curve prepared using indol−3-butyric acid (IBA—auxin equivalent) as the positive control at concentrations of 0.3–1 mg DW L^−1^.

#### 2.4.4. Cucumber Cotyledon Root Formation Test

The cucumber (*Cucumis sativus* L.) cotyledon expansion bioassay was used to determine the auxin-like activity of microalgae [39]. For this bioassay, the freeze-dried biomass extracts were diluted to 2 g DW L^−1^ concentration. Ten cotyledons were excised from 3-day-old seedlings and incubated on Petri dishes in a darkroom (26 °C) for 5 days. The number of roots formed at the bases of the cotyledons was then counted and compared between the treatments with microalgae extract and the IBA solutions. The IBA—auxin equivalent was dissolved in 95% ethanol. Microalgae extracts and IBA solutions were applied on individual 6-cm filter paper disks. To each treated disk, 3 mL of distilled water (negative control) were added to the bottom of each Petri dish. The IBA concentrations of 0.3–1 mg DW L^−1^ were used as the positive control for comparison.

### 2.5. Statistical Analysis

The reported values are the means of individual samples measured in triplicate (n = 3) and the error bars represent analytical standard deviation (SD). Sigma Plot 11.0 was used to determine significant differences between treatments using one-way *analysis of variance* (ANOVA) and the Holm–Sidak test for data evaluation. *p* values lower than 0.05 (*p* < 0.05) were considered significantly different.

## 3. Results

### 3.1. Cultivation Trials

The trials were carried out in mid-August 2021 using the bioreactors RWP1 and RWP2 and the data was used for comparison of culture behavior in units of different sizes. At the start of the trials, the bioreactors RWP1 and RWP2 were filled with urban WW and left to be populated by sewage-born microalgae culture. The microscopic analysis showed that the dominant strain in the developed water bloom was the green microalga *Micractinium pusillum* (*Chlorellaceae*) forming spherical cells with or without long spines (cell diameter of about 7 µm with biovolume of about 265 µm^3^), alternatively also grouping in colonies (Figure 2).

The bioreactors were operated in a continuous mode adding urban WW (dilution rate between 0.2–0.25 day^−1^; retention time of 4–5 days) from the nearby WWTP. The biomass density of the cultures was maintained between 0.44–0.44 g DW L^−1^ of which about 65–75% was microalgae/bacteria biomass. Thus, the biomass concentration ranged between 200 and 300 g DW m^−3^. The biomass productivity of the cultures was estimated between 15 and 25 g DW m^−2^ day^−1^. In a continuous system, it is complicated to estimate the difference in biomass concentration between the RW1 and RW2. The removal rate of nutrients from WW in the bioreactors was between 72 and 75% and 36 and 64% for nitrogen and phosphorus, respectively (Figure 3). In these trials, a N/P consumption ratio was 7 and 11 in the RWP1 and RWP2, respectively.

All days were hot and sunny with a clear sky and about 13 h of daylight. Irradiance data was recorded automatically using a data logger and a horizontal cosine-corrected sensor. The ambient maxima of about 1800 µmol photons m^−2^ s^−1^ were measured at 14:30 h and about 200 µmol photons m^−2^ s^−1^ at 9:00 h; the daily average was calculated between 500 and 570 µmol photons m^−2^ s^−1^.

During the cultivation trials, the ambient temperature maxima were between 40 and 45 °C. In the cultures, the morning temperature minima were about 23–24 °C while in the afternoon (17:00 h) maxima were between 31–32 °C (Table 2). The average values in bioreactors were about 28.5 °C which is the suitable temperature for green microalgae populations. The pH values varied in the range between 7.9 and 9.1 and 8.3 and 9.6 in RWP1 and RWP2, respectively (Table 2). It is important to note that the slightly lower average values (pH 8.6) were found in the large RWP1 compared to the RWP2 (pH 9.1).

### 3.2. Photosynthesis Measurements

The DO concentration in raceway ponds found in the morning (9:00 h) ranged between 4 and 19%sat, as a result of both microalgae and bacteria respiration (Table 2). At midday, the DO concentration increased to about 200 %sat due to photosynthetic oxygen production. The maxima of the DO concentration between 288–384 %sat were measured in the afternoon (17:00 h), as gas exchange with the environment was not sufficient due to relatively low culture turbulence. Aerobic and anoxic periods demonstrate the symbiotic interaction between microalgae and bacteria in the bioreactors. Even with the high photosynthetic activity, the maximum pH reached was 9.6. The photosynthetic activity might increase the pH value up to 11, but in this case, the additional CO_2_ is naturally provided by the bacteria.

In both bioreactors RWP1 and RWP2, the data of diurnal changes of photosynthetic activity measured in-situ showed very similar trends of diurnal changes in the irradiance inside the culture, relative electron transport rate rETR, and non-photochemical quenching NPQ (Figure 4). The illumination of the bioreactors was about 13 h long, between 8:30 and 21:30 h. The morning (9:00 h) irradiance intensities measured in the photic zone of the cultures were between 30–50 µmol photons m^−2^ s^−1^ which is close to the respiration/photosynthesis compensation point as estimated from light-response curves of oxygen production (data not shown). At midday (between 12:00 and 15:00 h), the irradiance maxima of about 200–250 µmol photons m^−2^ s^−1^ were found in the cultures (Figure 4). These intensities induced photosynthetic activity resulting in the rETR maxima between 20–25 μmol e^−^ m^−2^ s^−1^ (Figure 4, panels B). The NPQ values during the day were relatively stable between 0.25 and 0.55 in both bioreactors while at night the values were maximally up to 0.15 (Figure 4; panels C), most likely due to cell respiration. The daily course of rETR correlated with the course of ambient irradiance intensity penetrating the culture. It is important to emphasize here that in-situ measurements showed very similar physiological behavior of the cultures in both bioreactors.

Ex-situ measurements of RLC of Chl fluorescence were carried out in the samples taken at 9:30, 13:00, and 17:00 h on Days 2 and 3 from the bioreactors RWP1 and RWP2. The measurements of the Fv/Fm in both bioreactors revealed values between 0.62 and 0.71 which were relatively stable during the day (Figure 5A). In the morning, the rETRmax data were between 205 and 320 μmol e^−^ m^−2^ s^−1^ while at midday it ranged between 286 and 456 μmol e^−^ m^−2^ s^−1^ (Figure 5B). In the afternoon, the maximum values rETRmax were similar (between 300 and 410 μmol e^−^ m^−2^ s^−1^) to those measured at midday

The measurements of the Fv/Fm in all bioreactors revealed figures between 0.67 and 0.7, the data in both bioreactors was similar and only insignificant fluctuations were observed during the day (Figure 6A). The maximum values of rETRmax ranged between 310 and 440 μmol e^−^ m^−2^ s^−1^ during the day (Figure 6B). The values in the RWP1 and RWP2 were similar. Evaluation of photosynthesis light-response curves measured ex-situ revealed the Pmax values in all cultures mostly between 450 and 550 μmol O_2_ mg^−1^ Chl h^−1^ (Figure 6C). As concerns respiration rates, the values were found between 30–100 μmol O_2_ mg^−1^ Chl h^−1^. The highest respiration was found in the morning (9:00) (Figure 6D). Higher rETR, oxygen production, and respiration rates might be correlated with higher pH and irradiance values measured in the cultures in the bioreactors (Table 2, Figure 6C,D). Nevertheless, according to these data, all examined cultures were found in good physiological conditions, showing similar photosynthetic activity, thus assuming that the microalgae consortiums were similar in both bioreactors.

The Fv/Fm and rETRmax values estimated from rapid and slow (classical) light-response curves of Chl fluorescence provided comparable data and showed similar trends (Figure 5A,B and Figure 6A,B) even if time steps were quite different, i.e., 10 s vs. 2 min. The only difference might be a higher scatter of values, probably too short exposure intervals of individual light intensities of RLC (Figure 5B). It is worth pointing out that rETRmax values measured in samples in the laboratory were one order of magnitude higher (Figure 6B) than those measured in the cultures in-situ (Figure 4B). This difference is caused by the much higher light intensity used to obtain maximum ETR in the laboratory samples.

The culture samples taken from the bioreactors were examined by the OJIP test to estimate the redox status of quinone electron acceptors in the PSII complex (Figure 7). The course of the curves in the samples taken at 9:00 and 13:00 h showed very similar trends in RWP1 and RWP2. Only in the afternoon (17:00 h) were the J and I inflection points on the induction curve slightly increased, which might suggest some temporary slow-down of the electron transport on the reducing side of the PSII complex (Figure 7D,E).

### 3.3. Bioactivity Measurements

The auxin- and cytokinin-like bioactivities were examined in water extracts of freeze-dried microalgae samples. The auxin-like activity corresponding to 0.3–0.7 mg IBA L^−1^ (standard) was found in the microalgae cultures taken from the bioreactors when the cucumber cotyledon root development test was used (Figure 8A). The auxin-like bioactivity was found between 113% and 136% of the negative control (water) in the samples taken from RWP1 and RWP2. However, only one sample showed auxin-like activity when the mung bean root development test was used (Figure 8B).

When the cytokinin-like activity was examined in the microalgae samples collected from outdoor bioreactors, some activity was found when using the leaf chlorophyll retention test, namely 116–164% of the control (Figure 8C,D). Nevertheless, only the biomass samples in Trial 2 taken from the RWP2 bioreactor were statistically different from the control but remained below the chlorophyll value of all tested IBA concentrations. The cucumber cotyledon expansion test revealed no cytokinin-like activity in microalgae samples (Figure 8D).

## 4. Discussion

Populations of microalgae that naturally develop in open reservoirs like raceway ponds after filling with municipal WW are dominated mostly by green microalgae. In such cultivation units, the microalgae population that appeared during WWT mostly consisted of *Micractinium* sp. and *Desmodesmus* sp. as reported previously [12]. The growth of microalgae cultures depends on their photosynthetic activity. If the conditions for microalgae culturing in outdoor bioreactors are considered, the crucial variables governing the photosynthetic activity and subsequent growth of microalgae mass cultures are irradiance intensity, temperature, pH, CO_2_/O_2_ exchange, nutrient supply, culture turbulence, biomass density, and culture depth (light path) [42,43]. In large-scale microalga cultures, warning signals have to be recognized as soon as possible to prevent a significant reduction in daily productivity, or to avoid situations that, in a few days, may culminate in culture loss.

In the present trials, various fast photosynthesis monitoring techniques have been employed [27] to find out whether cultures grown in municipal WW and high irradiance may experience transient stress and validate photosynthesis and fluorescence measurement techniques in large-scale bioreactors in Mérida. The ambient irradiance maxima of about 1800–2000 μmol photons m^−2^ s^−1^ is usually available on clear summer days which is about 10-times higher intensity than that required to saturate photosynthesis. The measurement of light intensity in the photic layer in microalgae cultures showed a mean light intensity of about 200 μmol photons m^−2^ s^−1^, which is considered an optimum of saturating irradiance for green microalgae growth [44,45]. In the present trials, the photosynthetic activity was measured in-situ in the photic zone of the bioreactors (Figure 4), where the measured irradiance intensity was similar. The Fv/Fm variable is often used as an estimate of the photochemical yield of PSII and its decrease usually indicates that the cultures are exposed to unfavorable conditions [43]. The Fv/Fm values around 0.7 also showed that the welfare of microalgae cultures was relatively good. Similar values were found in the healthy-growing outdoor cultures of *Chlorella* [45]. Data gathered by photosynthesis measurements indicated that cultures maintained a stable Fv/Fm ratio even at midday when irradiance intensity reached the maximum. Further confirmation of optimal physiological condition was attained with in-situ ETR measurements, which followed the course of light intensity without any sign of photoinhibition/downregulation of the PSII complex at midday, as well as low induction of NPQ remaining rather low during the day. It appeared that NPQ did not relax completely even at night, the cause of which is very likely to be high respiration rate of cells maintaineing the thylakoids energized. The increase in the size of the pond from 0.5 ha to 1 ha did not affect the validity of fast measurement techniques, which showed similar behavior.

Generally, the acceptable growth temperature for most microalgae species ranges between 15 and 35 °C and temperature optima are close to 30 °C [45]. Under suboptimal temperatures, growth rates are low while the superoptimal values may inhibit growth. In such a temperature range, photosynthesis in most microalgae species becomes saturated at light irradiances of 200–250 μmol photons m^−2^ s^−1^ [44,45].

The pH values in the cultures roughly indicate CO_2_ availability for growth [46,47]. In outdoor bioreactors, CO_2_ might be supplied based on pH measurements keeping the value around 8 [8,48,49]. Without CO_2_ addition, the photosynthetic activity causes an increase in the pH value up to 11 which may inhibit both microalgae and bacteria activity [50]. In the present outdoor trials, only ambient CO_2_ and that produced by bacteria in bioreactors were available; no additional CO_2_ was supplied to the bioreactors. Thus, pH values increased during microalgae cultivation from morning to afternoon measurements, as a result of a partial CO_2_ deprivation (Table 2). Nevertheless, photochemical activity measured as a build-up of DO concentration and rETR activity in-situ was not significantly limited (Figure 4 and Figure 5) as the cultures were rather diluted.

Dissolved oxygen concentrations usually reflect the diurnal cycle of irradiance intensity showing the build-up from morning minima to irradiance maxima at midday and then the values decline through the afternoon. The build-up in the DO concentration due to photosynthetic activity was generally accompanied by an increase in pH due to CO_2_ uptake [27]. Under high photosynthesis rates, DO concentration, can reach up to 25–30 mg DW L^−1^ (200–400% of air saturation) at the top layer of microalgae cultures even in open bioreactors [30,45]. High DO concentrations, particularly in synergism with high irradiance may impact growth due to the potential slow-down of photochemical yield [49]. Excessive DO concentration may cause increased photorespiration which can significantly reduce productivity [51]. However, green microalgae use non-photochemical quenching to dissipate excess energy via the xanthophyll cycle [52].

The Chl fluorescence data and oxygen production measured online/in-situ demonstrated a prompt response of the cultures as the increase of rETR and rapid build-up of DO concentration from the morning towards midday and afternoon as well as high values of the maximum photochemical yield of PSII, Fv/Fm measured ex-situ during the day. The kinetics of fast fluorescence induction also did not show any significant electron transfer disturbance in the PSII complex in both bioreactors (Figure 7) as demonstrated for some microalgae [27]. Otherwise, the course of curves was similar to the measurements taken in the healthy cultures of green microalgae.

According to the present trials, the data demonstrated that the microalgae were in decent ‘health‘, showing good areal biomass productivity if we consider that the cultures were grown only on ‘ambient’ CO_2_. Although the microalgae culture and wastewater sources were the same in both bioreactors, a certain difference in nutrient consumption might be affected by the mixing, volume, temperature, and light availability of the bioreactor. The efficiency of the microalgae–bacteria consortia for nutrient uptake is not only affected by the bioavailability of nutrients, but also depends on the complex interactions between physicochemical factors, such as pH, light intensity, photoperiod, temperature, and biological factors [53]. Thus, the nutrient removal in these systems depends on their assimilation by the microalgae, biological processes (nitrification/denitrification), and other phenomena, such as ammonia volatilization, and phosphorus precipitation. The removal rate of nutrients from WW by the microalgae population in the presented trial was up to 75% and 64% for nitrogen and phosphorus, respectively (Figure 3). A higher average pH value may support the removal activity as the mechanism of phosphorus removal via the assimilation by the microalgae and bacteria (and some precipitation). For example, the values were comparable to *Chlorella* cultures, showing a removal efficiency of about 85% of phosphorous content and about 89% of nitrogen contents [54].

One of the ways to explore microalgae biomass is the production of various biofuels providing low-cost raw biomass. In one DEMO study [11], the design and operation of a 5-ha raceway bioreactor was operated at the Christchurch WW treatment plant (New Zealand), and the produced microalgae–bacteria biomass was used as an energy source for local communities.

Recently, ways to improve agricultural yields have been subject to much research. Over the last two decades, natural plant stimulants (biofertilizers and biostimulants) have received increasing attention from the scientific community and agrochemical industries to improve yields and sustainability by replacing agrochemicals (synthetic fertilizers, herbicides, and pesticides) whose residues lead to environmental pollution [55,56]. The biostimulant market has been one of the fastest-growing agriculture-related sectors (EU 2019/1009). These naturally produced compounds are a variety of biologically active molecules that can positively affect plant growth as well as the efficiency of nutrient use and increase plant tolerance to abiotic stresses. In this respect, microalgae strains showing biostimulating features have been studied as these microorganisms can be produced in controlled mass cultures and their metabolic plasticity provides the possibility to modulate biomass composition which is one of the critical aspects in the production of biostimulants [57,58,59,60,61,62]. Recently, the pilot production trial of *Scenedesmus* and *Chlorella* biomass with biostimulating features was realized using brewery [63] or municipal WW [37,64] as sources of nutrients which significantly reduced the costs of biomass production. Biotests also showed that *Micractinium* is a valuable strain with biostimulating activity for agricultural use based on its auxin-like activity [61]. Downstream processes, e.g., dissolved air flotation, centrifugation, sonication, and enzymatic hydrolysis, are being optimized to prepare biomass extracts in DEMO scale for field or greenhouse hydroponic application.

## 5. Conclusions

In the present large-scale trials, microalgae cultures (strain *Micractinium* sp. which naturally developed in urban WW) were grown in large-scale bioreactors located in Mérida (Spain). These were operated in continuous régime (0.2–0.25 d^−1^; retention time of 4–5 days) using municipal WW as the sole water and nutrient source. Neither CO_2_ nor air was bubbled into the culture. Yet, the cultures grew well in both bioreactors resulting in biomass productivity of 15–25 g DW m^−2^ day^−1^. Photosynthesis monitoring in-situ and ex-situ revealed relatively good photosynthetic activity of the cultures. The biomass extracts showed biostimulating activity.

The important output of these trials was the validation of measuring techniques for the monitoring and optimization of microalgae growth in large-scale bioreactors. Photosynthesis monitoring for oxygen production and Chl fluorescence were proven as a suitable tool showing the physiological status of a culture and can be used to adjust suitable growth regimes and estimate biomass production in large-scale bioreactors. This work has been a valuable study on the actual physiological status and behavior of microalgae populations in large-scale plants used for municipal WW remediation.

## Figures and Tables

**Figure 1 biology-11-01380-f001:**
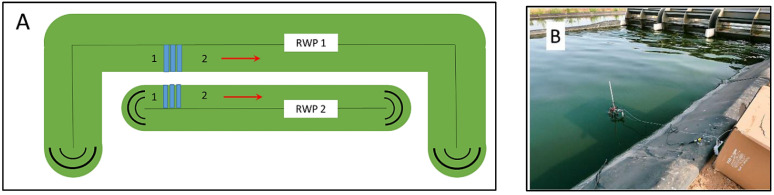
(**A**) Schematic diagram of raceway ponds RWP1 and RWP2 in the DEMO 5 plant constructed in Mérida, Spain (GPS coordinates 38°55′52.1″ N, 6°23′35.7″ W). Points 1 and 2 indicate where pH, dissolved oxygen (DO) concentration, and temperature were measured (1—before paddle wheel; 2—after paddlewheel). Data was collected in bioreactors RWP1 and RWP2. (**B**) In-situ Chl fluorescence in microalgae cultures grown in bioreactors was measured using a portable fluorimeter (Junior-PAM fluorimeter, H. Walz, Effeltrich, Germany). The irradiance sensor and Chl fluorescence fiberoptics were submerged directly in the culture side by side in the photic zone (about 15 mm deep) in position 1 before the paddlewheel.

**Figure 2 biology-11-01380-f002:**
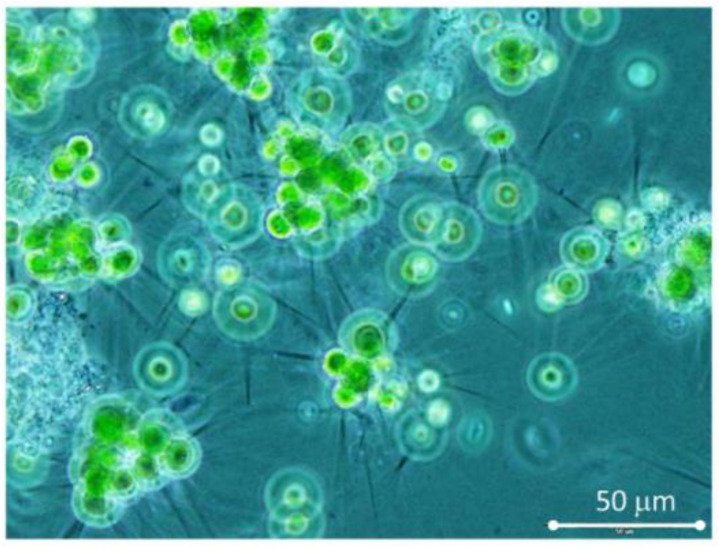
Spherical single cells or colonies of the green microalgae *Micractinium pusillum* with and without long spines (cell of about diameter 8 µm, biovolume 265 µm^3^) were the major species in the water bloom grown in bioreactors. The image was taken at 400× magnification using phase contrast illumination.

**Figure 3 biology-11-01380-f003:**
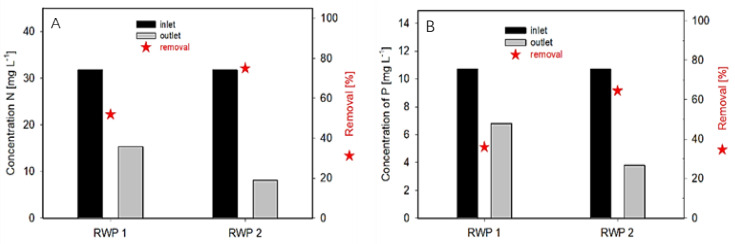
Nitrogen (**A**) and phosphorus (**B**) content in municipal WW and removal efficiency by microalgae cultures at the output in the bioreactors RWP1 and RWP2. The WW (inlet) and microalgae culture (outlet) were supplied/harvested continuously over the day and then averaged samples were analyzed for bioreactors RWP1 and RWP2.

**Figure 4 biology-11-01380-f004:**
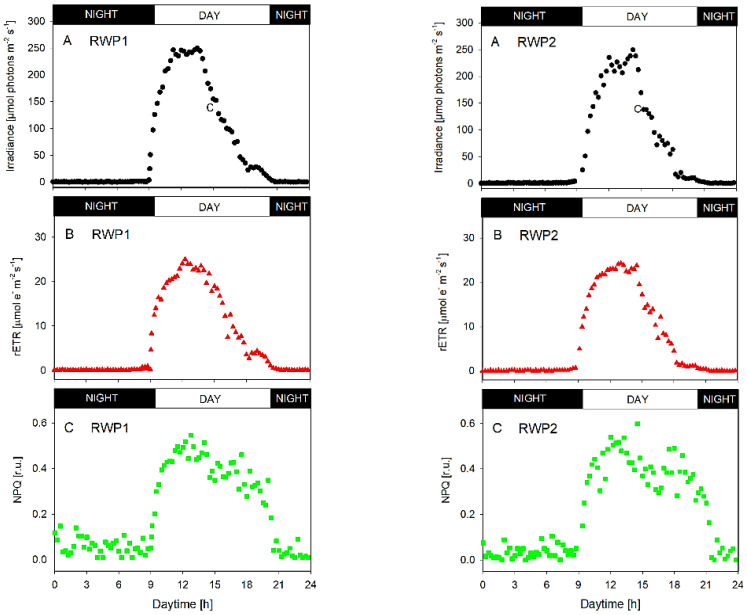
Diurnal changes of irradiance and photosynthetic activity (estimated in-situ/online in the photic zone) in the microalgae culture grown outdoors in the RWP1 (**left panels**) and RWP2 (**right panels**) bioreactors. (**A**) Irradiance intensity in the culture measured in the photic zone (the depth of light penetration); (**B**) relative electron transport rate rETR = [YII × PAR]; (**C**) non-photochemical quenching NPQ.

**Figure 5 biology-11-01380-f005:**
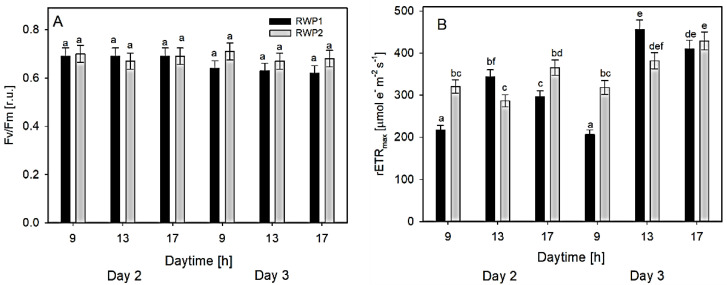
Variables Fv/Fm (**A**) and rETRmax (**B**) were estimated ex-situ using rapid light-response curves of Chl fluorescence. The column designation is identical in both panels. The data were measured in the culture samples taken at 09:00, 13:00, and 17:00 h from the bioreactors RWP1 and RWP2 on days 2 and 3. The mean values designated by the same letter did not differ significantly from each other.

**Figure 6 biology-11-01380-f006:**
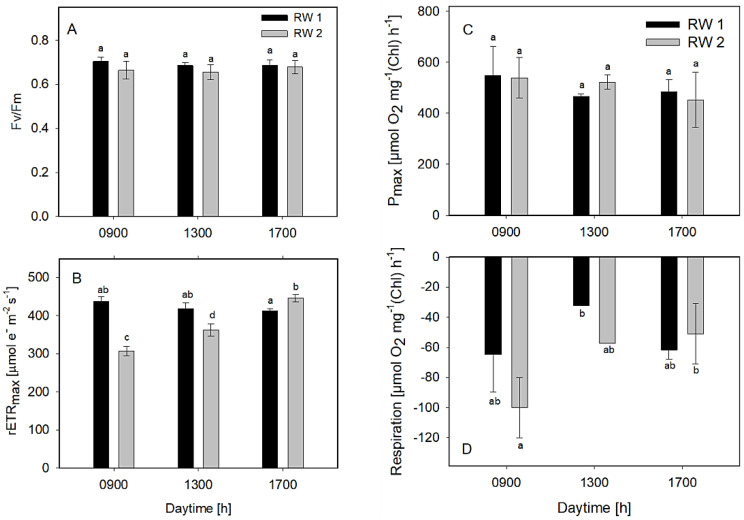
Variables Fv/Fm (**A**), rETR (**B**), oxygen production (**C**), and respiration (**D**) were estimated ex-situ using slow light-response curves of photosynthesis. The column designation is identical in all panels. The data were measured in the culture samples taken at 09:00, 13:00, and 17:00 h from the bioreactors RWP1 and RWP2 during outdoor trials in Mérida. The mean values designated by the same letter did not differ significantly from each other.

**Figure 7 biology-11-01380-f007:**
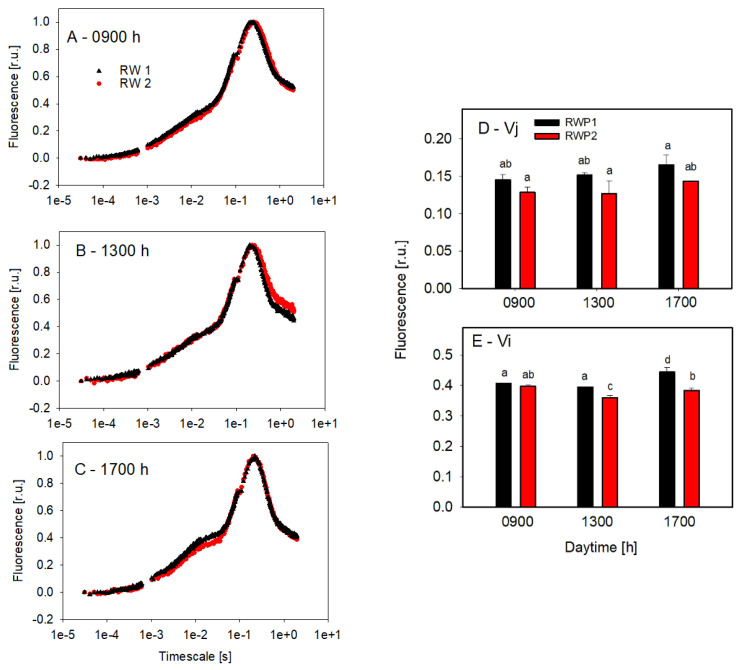
Curves of fast fluorescence induction kinetics measured ex-situ in the culture samples taken at 09:00 h (**A**), 13:00 h (**B**), and 17:00 h (**C**) from the bioreactors RWP1 and RWP2 during outdoor trials in Mérida. Each curve is an average of 3–5 measurements which were double normalized to the minimum (F_0_) and maximum points (Fm). The curve designation is identical in panels (A–C) and the column designation is the same in panels (D,E). Variables Vj (**D**) and Vi (**E**) were estimated from fast fluorescence induction curves. The mean values designated by the same letter did not differ significantly from each other.

**Figure 8 biology-11-01380-f008:**
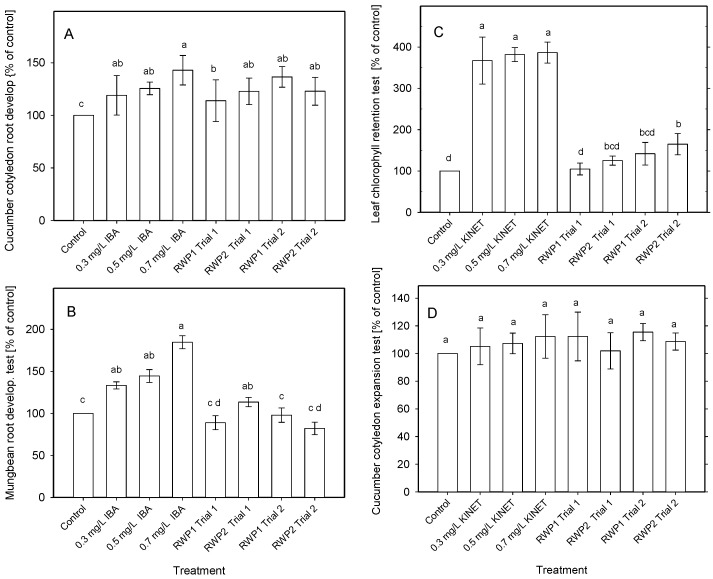
The auxin activity of microalgae was detected using the cucumber cotyledon root development test (**A**) or by mung bean root development test (**B**). Indol−3-butyric acid (IBA) was used as the standard. The cytokinin-like activity was detected using the leaf Chl retention test (**C**), and the cucumber cotyledon expansion test (**D**). Kinetin was used as the standard (positive control). Water extracts of freeze-dried microalgae samples (2 mg DW L^−1^) were taken from the bioreactors RWP1 and RWP2, respectively on Days 1 and 3 during outdoor trials in Mérida. The results are presented as mean ± SD (*n* = 18). Different letters indicate significant differences analyzed by one-way analysis of variance and Duncan’s multiple range test (*p* < 0.05).

**Table 1 biology-11-01380-t001:** Averaged composition of wastewater (COD—chemical oxygen demand, nitrogen, and phosphorous content) supplied to outdoor bioreactors RWP1 and RWP2 during the outdoor trial and nutrient removal rate. The WW (inlet) and microalgae culture (outlet) were supplied/harvested continuously over the day and then averaged samples were analyzed for bioreactors RWP1 and RWP2.

	COD (mg DW L^−1^)		Nitrogen (mg DW L^−1^)		Phosphorous (mg DW L^−1^)	
	RWP1	RWP 2	RWP1	RWP 2	RWP1	RWP 2
**Inlet WW**	550	550	58	58	10.7	10.7
**Outlet culture**	82	73	16	12	6.8	3.8
**Removal (%)**	85	85	72	79	36	65

**Table 2 biology-11-01380-t002:** Measurements of temperature, pH, and dissolved oxygen (DO) concentration were carried out in the RWP1 and RWP2 bioreactors at 9:00, 13:00, and 17:30 h on days 2 and 3 during outdoor trials. The data are discrete measurements of variables taken after the paddlewheel (position 1 in Figure 1A) as there was little variation along the bioreactors.

Day	Daytime	Temperature (°C)		pH		DO Concentration (% of Saturation)	
		RWP1	RWP2	RWP1	RWP2	RWP1	RWP2
Day 2	9:00	24.4	23.8	8.0	8.7	12	14
	13:00	27.6	27.7	8.7	9.2	198	204
	17:00	31.9	32.2	9.0	9.6	288	319
Day 3	9:00	25.2	24.3	7.9	8.3	16	19
	13:00	28.9	28.9	8.8	9.0	221	197
	17:00	33.3	33.8	9.1	9.6	325	278

## Data Availability

The data presented in this study are available from the corresponding author upon reasonable request. The datasets are not publicly available without the permission of all co-authors.

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
