# Peer review of "Photosynthesis Monitoring in Microalgae Cultures Grown on Municipal Wastewater as a Nutrient Source in Large-Scale Outdoor Bioreactors"

_biology, 2022, doi:10.3390/biology11101380_

Round 1
Reviewer 1 Report
Dear editor, dear authors,
Here is my review of the article biology-1899245 entitled "Photosynthesis Monitoring in Microalgae Cultures Grown on Municipal Wastewater as a Nutrient Source in Large-scale Outdoor Bioreactors".
The manuscript presents large-scale outdoor algal bioreactors using water and nutrients from municipal wastewater in Mérida, Spain. The authors determined the predominant algal strain and its productivity. The authors implemented and validated different measurement techniques to monitor and optimize microalgae growth. They determined the growth of the culture, monitored the photosynthetic activity in-situ and ex-situ, and the physiological status of the microalgae culture.
The results show suitable growth regimes for biomass production in large-scale bioreactors. They showed good growth and behavior of microalgae populations. Analysis of microalgal biomass showed its potential for agricultural use as biofertilizer. The bioassays to determine the activities of plant growth regulators revealed activities of plant growth regulators that can potentially be used as biostimulants.
General recommendations:
The research topic on both subjects is suitable for publication, but the manuscript has some shortcomings related to the written design of the subject. In my opinion, it needs improvement and require a minor revision!
Comments to authors:
Short summary:
The first sentence is not necesrarry.
Abstract.
Sentences need to be shortened: "The objective was to validate measurement techniques for monitoring culture performance in large-scale bioreactors. Different in situ/ex situ photosynthesis measurement techniques, namely oxygen production and chlorophyll fluorescence, were used to characterize the physiological behavior and growth of the cultures."
Introduction:
Is too long.
1st paragraph:
“Sentence “For example, using Chlorella cultures with a high removal ef-78 ficiency of about 85% of phosphorous content and about 89% of nitrogen contents 79 were reported [8]. needs to be included in the discussion.
The data on commercial production at introduction differ from the data in the discussion. This paragraph is too long and superficial.
There is no calculation of costs in the results. If the authors mention production costs in the introduction, they need to calculate them in the results and then discuss them in the discussion or delete everything from the manuscript!
The paragraph from line 133 to 141 is not necessary for the manuscript. Please omit it from the introduction.
The purpose is missing and several goals must be present (see the difference). Some of them were not even mentioned.
Material and methods:
General comment is that several methods are cited with to many references. Please specify which reference is for particular part of the method.
Please explain calculation of NPQ!
Explain what is Pmax?
Please explain OJIP: “Fast fluorescence induction kinetics” (Kautsky curve or OJIP test)
Analitic measurements can be converted into biomass determination and biochemical measurements.
Bioassays can be converted into "Determination of biological activity".
Please check the references related to bioassays. Some of them are listed for reference only, and some of them do not allow replication of the methods.
Results:
The figures are not self-explanatory. All figures and tables should be self-explanatory, i.e., the figure and table headings must include the number of explants. A description of the statistics is missing. Which statistical program was used, what was SD or SE? Which post-hoc test was used?
According to the data, there is no citokinin-like biological activity - sorry. (Figure 8).Results:
Figures are not self-explanatory. All figures and tables should be self-explanatory, so figure and table captions must include the number of explants. There is a lack of description of statistics. What statistical program was used, what was SD or SE? What post-hoc test was used?
According to data there is no citokinin–like biological activity- sorry. (Figure 8).
Discussion
The results do not include a calculation of costs. If the authors mention production costs in the introduction, they must calculate them in the results and then discuss them in the Discussion or remove them from the manuscript altogether!
Please reconsider paragraph 595-608 and link it to the results or omit it from the discussion.
Please reconsider paragraphs 608-615 It is not connected to the results. Please leave it out of the discussion.
Please reconsider paragraphs 618-626: Too many references and again no connection to the results. Please remove it from the discussion.
Please reconsider paragraphs 626-632: Too superficial, no connection with auxin-like activity of algae. Please reconsider this discussion.
Reconsider conclusion after recommendations of results and discussion.
Author Response
Response to the Comments of Reviewer 1
General recommendations: The research topic on both subjects is suitable for publication, but the manuscript has some shortcomings related to the written design of the subject. In my opinion, it needs improvement and requires a minor revision!
Comments to authors:
Comment: Short summary: The first sentence is not necessary.
Response: The sentence was deleted.
Comment. Abstract - Sentences need to be shortened. The text was revised: ‘The objective was to validate measurement techniques for monitoring culture performance in large-scale bioreactors. Different in situ/ex situ photosynthesis measurement techniques, namely oxygen production and chlorophyll fluorescence, were used to characterize the physiological behavior and growth of the cultures.’
Response: This part was rephrased as follows: The objective was to validate photosynthesis monitoring techniques in large-scale bioreactors. Various in‑situ/ex‑situ methods based on chlorophyll fluorescence and oxygen evolution measurements were used to follow culture performance. Photosynthesis variables gathered with these techniques were compared to the physiological behavior and growth of the culture
Comment: Introduction - Is too long.
Response: As requested the Introduction part was significantly shortened.
Comment: 1st paragraph: Sentence ‘For example, using Chlorella cultures with a high removal efficiency of about 85% of phosphorous content and about 89% of nitrogen contents were reported [8]’ needs to be included in the discussion.
Response: The sentence was included in the text of the Discussion (p. 16).
Comment: The data on commercial production in the introduction differs from the data in the discussion. This paragraph is too long and superficial.
Response. This part of the Introduction was significantly reduced and revised (p. 3).
Comment: There is no calculation of costs in the results. If the authors mention production costs in the introduction, they need to calculate them in the results and then discuss them in the discussion or delete everything from the manuscript!
Response: The matter of cost calculation was omitted from the text as it is out of the scope of the MS topic.
Comment: The paragraph from line 133 to 141 is not necessary for the manuscript. Please omit it from the introduction.
Response: This part of the text was removed from Introduction and partly used in the 1st paragraph of the Discussion (p. 14-15).
Comment: The purpose is missing and several goals must be present (see the difference). Some of them were not even mentioned.
Response. The last paragraph of the Introduction was rephrased: ‘In this work, photosynthesis measuring techniques were validated for monitoring microalgae cultures grown in outdoor bioreactors which were used for municipal wastewater remediation. Photosynthetic activity measured by oxygen production and Chl fluorescence -in-situ/ex-situ was found as a suitable tool to follow the physiological status and can be used to estimate the growth regime of microalgae cultures in large-scale bioreactors.’
Comment: Material and methods - General comment is that several methods are cited with too many references. Please specify which reference is for a particular part of the method.
Response: The number of references concerning the use of some techniques in Materials and Methods was reduced to the substantial ones, maximally to two.
Comment: Please explain calculation of NPQ!
Response: The sentence on NPQ calculation was added in the part of Methods - Ex-situ measurements.
Comment: Explain what is Pmax?
Response: The sentence on Pmax calculation was added in the part of Methods - Ex-situ measurements.
Comment: Please explain OJIP: “Fast fluorescence induction kinetics” (Kautsky curve or OJIP test)
Response: The text explaining the OJIP test was added in the part of Methods - Ex-situ measurements.
Comment: Analytic measurements can be converted into biomass determination and biochemical measurements.
Response: The paragraph title was modified to Biomass determination and biochemical measurements
Comment: Bioassays can be converted into "Determination of biological activity".
Response: The subtitle was changed.
Comment: Please check the references related to bioassays. Some of them are listed for reference only, and some of them do not allow replication of the methods.
Response: The list of references was controlled and revised.
Comment: Results - The figures are not self-explanatory. All figures and tables should be self-explanatory, i.e., the figure and table headings must include the number of explants (????). (maybe he wants the number of replicates, n=3) There is a lack of description of statistics. What statistical program was used, what was SD or SE? What post-hoc test was used?
Response: Legends of most figures and tables were revised; in most of them the text was extended to be more self-explanatory.
As concerns replicates – Table 1 and Table 2 show averaged or discrete measurements of variables in-situ as there was little variation along the bioreactors. In Figure 3, the WW and microalgae cultures were supplied/harvested at the inlet/outlet continuously over the day, and then averaged samples were analyzed for bioreactors RWP1 and RWP2. Statistics were considered unnecessary.
Figure 4 shows a series of continuous measurements carried out online every 10 min, statistical evaluation of the data does not have any sense.
In Figure 7A each curve is an average of 3-5 almost identical curves; statistical evaluation would be superfluous; moreover, error bars might ‘mess’ the curves.
In Figures 5, 6, 7A, and 8 the data are presented as mean ± SD (n=3) and were analyzed treatments using one-way analysis of variance (ANOVA) and the Holm-Sidak test or Duncan’s multiple range test (P < 0.05).
Comment: According to the data, there is no cytokinin-like biological activity - sorry. (Figure 8).
Response: We agree; it was not correctly referred. The text in the Results (p. 16) part was revised to: ‘When the cytokinin-like activity was examined in the microalgae samples collected from outdoor bioreactors some activity was found when using the leaf chlorophyll retention test – 116-164 % of the control (Figures 8C, D). Nevertheless, only the biomass samples in Trial 2 taken from the RWP2 bioreactor were statistically different from the control but remained below the chlorophyll value of all tested IBA concentrations. The cucumber cotyledon expansion test revealed no cytokinin-like activity in microalgae samples (Figure 8D).’
Comment: Discussion - The results do not include a calculation of costs. If the authors mention production costs in the introduction, they must calculate them in the results and then discuss them in the Discussion or remove them from the manuscript altogether!
Response: As mentioned earlier in the text, the estimate/calculation of costs was omitted from the text as it was not a matter of the trial and it is out of the scope of the MS purpose.
Comment: Please reconsider paragraph 595-608 and link it to the results or omit it from the discussion.
Response: The part of this paragraph was omitted from the Discussion. Yet, we consider it important to mention large-scale facilities for the production of low-cost biomass.
Comment: Please reconsider paragraphs 608-615 It is not connected to the results. Please leave it out of the discussion.
Please reconsider paragraphs 626-632: Too superficial, no connection with auxin-like activity of algae. Please reconsider this discussion.
Response: The last paragraph of the Discussion was substantially revised; the part was also omitted. The number of literature references was reduced from 78 to 64. Yet, we consider as important to mention the agricultural importance and use of biostimulants in the Discussion part.
Comment: Reconsider conclusion after recommendations of results and discussion.
Response: The Conclusion part was significantly revised to reflect the purpose of this work.
Reviewer 2 Report
In the discussions, the problem of the different absorption of phosphorus depending on the volume of the bioreactor is not present. What would be the cause of this phenomenon and a possible impact on the pH of the environment, the vital activity of microalgae being identical in both types of bioreactors;
In figures 5 and 6 to check the correctness of the notes on the x axis
Author Response
Response to comments of Reviewer 2
Comment: In the discussions, the problem of the different absorption of phosphorus depending on the volume of the bioreactor is not present. What would be the cause of this phenomenon and a possible impact on the pH of the environment, the vital activity of microalgae being identical in both types of bioreactors;
Response: One paragraph concerning this comment was added in the Discussion (p. 17).
‘According to the present trials, the data have demonstrated that the microalgae were in decent ‚health‘ status showing good areal biomass productivity if we consider that the cultures were grown only on ‘ambient’ CO2. Although the microalgae culture and wastewater sources were the same in both bioreactors, a certain difference in nutrient consumption might be affected by the mixing, volume, temperature, and light availability of the bioreactor. The efficiency of the microalgae-bacteria consortia for nutrient uptake is not only affected by the bioavailability of nutrients but also depends on the complex interactions between physicochemical factors such as pH, light intensity, photoperiod, temperature, and biological factors [54]. Thus, the nutrient removal in these systems depends on their assimilation by the microalgae, biological processes (nitrification/denitrification), and other phenomena such as ammonia volatilization and phosphorus precipitation. The removal rate of nutrients from WW by the microalgae population in the presented trial was up to 75% and 64% for nitrogen and phosphorus, respectively (Figure 3). A higher average pH value may support the removal activity as the mechanism of phosphorus removal via the assimilation by the microalgae and bacteria (and some precipitation). For example, the values were comparable to Chlorella cultures showing a removal efficiency of about 85% of phosphorous content and about 89% of nitrogen contents [55].
Comment: In figures 5 and 6 to check the correctness of the notes on the x-axis
Response: In Fig. 5 the description of the X-axis was modified.